

# Connecting virtual reality and ecology: a new tool to run seamless immersive experiments in R

Julie Vercelloni[1,2], Jon Peppinck[1,2], Edgar Santos-Fernandez[1,2], Miles McBain[1], Grace Heron[1], Tanya Dodgen[1], Erin E. Peterson[1,2] and Kerrie Mengersen[1,2]

[1] Queensland University of Technology, Australian Research Council, Centre of Excellence for Mathematical and Statistical Frontiers, Brisbane, Australia
[2] School of Mathematical Sciences, Faculty of Science and Engineering, Queensland University of Technology, Brisbane, Brisbane, QLD, Australia

## ABSTRACT

Virtual reality (VR) technology is an emerging tool that is supporting the connection between conservation research and public engagement with environmental issues. The use of VR in ecology consists of interviewing diverse groups of people while they are immersed within a virtual ecosystem to produce better information than more traditional surveys. However, at present, the relatively high level of expertise in specific programming languages and disjoint pathways required to run VR experiments hinder their wider application in ecology and other sciences. We present R2VR, a package for implementing and performing VR experiments in R with the aim of easing the learning curve for applied scientists including ecologists. The package provides functions for rendering VR scenes on web browsers with A-Frame that can be viewed by multiple users on smartphones, laptops, and VR headsets. It also provides instructions on how to retrieve answers from an online database in R. Three published ecological case studies are used to illustrate the R2VR workflow, and show how to run a VR experiments and collect the resulting datasets. By tapping into the popularity of R among ecologists, the R2VR package creates new opportunities to address the complex challenges associated with conservation, improve scientific knowledge, and promote new ways to share better understanding of environmental issues. The package could also be used in other fields outside of ecology.

Corresponding author
Julie Vercelloni,
j.vercelloni@qut.edu.au

## BACKGROUND

The emergence of digital technologies, including Virtual Reality (VR), facilitates connections between the public and the scientific community and creates innovative pathways for environmental conservation research (*Mazumdar et al., 2018*; *Queiroz et al., 2019*; *Fauville, Queiroz & Bailenson, 2020*). In general, VR uses a combination of immersive technology via head-mounted devices, hand controllers and stereoscopic sound to replace natural sensory input with inputs from a computer system, such that a person is exposed to vivid virtual scenes (*Riva et al., 2007*). In the field of ecology, VR experiences are used as

a research tool to (1) increase understanding about the complexity of environmental issues associated with climate change, (2) influence empathy, and (3) promote environmental behavior changes (*Markowitz et al., 2018*; *Herrera et al., 2018*; *Queiroz et al., 2019*; *Nelson, Anggraini & Schlüter, 2020*). Despite promising results, the small number of published studies that have used VR approaches in ecology shows that there remain opportunities for further research in environmental education (*Queiroz et al., 2019*; *Fauville, Queiroz & Bailenson, 2020*) and for the development of programming tools that ease the integration of VR with applied science fields (*Okamoto et al., 2012*; *Jangraw et al., 2014*; *Vasser et al., 2017*; *Loup et al., 2018*; *Brookes et al., 2019*; *Bexter & Kampa, 2020*).

VR experiments for environmental conservation involve the elicitation of information while people are immersed in virtual scenes of natural ecosystems, such as 360-degree images. VR experiments include multimodal features of text, images, sounds and haptic feedback to create a rich and engaging environment to expose people to more complete and complex information (*Fauville, Queiroz & Bailenson, 2020*).

In the fields of ecology and conservation, VR has the potential to support greater understanding of complex ecological processes such as coral bleaching (*Minocha, Tudor & Tilling, 2017*), and new forms of thinking about ecosystem dynamics (*Grotzer et al., 2015*; *Queiroz et al., 2019*). VR experiments solve the difficulty of accessing ecosystems that are situated in remote locations and might be potentially dangerous or expensive to survey. Continuous access to these ecosystems opens up new opportunities for ecologists to fill the gaps in current scientific knowledge related to the paucity of data and ecological consequences of major changes in ecosystems health and species composition.

Combined with modern techniques in statistical ecology, elicited information collected from VR experiments can produce new types of ecological insights that complement environmental monitoring and conservation efforts. For example, VR experiments with 360-degree images were used to develop aesthetic indicators based on people's perception of the beauty of a coral reef (*Vercelloni et al., 2018*). They were also used to predict the presence or absence of emblematic species threatened by habitat loss and fragmentation, such as koalas (*Phascolarctos cinereus*, (*Leigh et al., 2019*), Australian rock wallabies (*Petrogale penicillata*, (*Brown et al., 2016*), and jaguars (*Panthera onca*) (*Bednarz et al., 2016*; *Mengersen et al., 2017*). In these experiments, opinions and knowledge were extracted from the responses given by experts, indigenous communities, scuba-divers and non-expert participants. This information was then incorporated into quantitative statistical models and used to improve understanding of complex ecological systems and to inform the development of future management and conservation strategies. Such strategies included the creation of a jaguar conservation corridor across the Amazon rainforest (*Zeller et al., 2013*) and supporting the Australian government in their reporting to UNESCO on the status of the Great Barrier Reef World Heritage Area (*Vercelloni et al., 2018*).

VR experiments in ecology are often conducted using generic VR experiences such as Google Expeditions or pre-made 360-degree movies (*McMillan, Flood & Glaeser, 2017*; *Parmaxi, Stylianou & Zaphiris, 2017*; *Nelson, Anggraini & Schlüter, 2020*), which are primarily developed for educational purposes (*Markowitz et al., 2018*). These tools are not designed to be adapted for specific research purposes, therefore a collaboration with

VR developers and accessibility to bespoke VR software is required to repurpose them for research applications (*Loup et al., 2018*). Common VR programming environments such as C#/Unity (https://unity3d.com), C++/Unreal Engine (https://www.unrealengine.com/en-US/) and React 360 (https://opensource.facebook.com/) require specific programming expertise, which ecologists and other scientists may lack.

The R2VR package development was motivated with the goal of providing greater access to VR experiments and the associated research benefits of using the R statistical software environment, a top ten popular programming language (https://www.tiobe.com/tiobe-index/) extensively used by quantitative ecologists (*Lai et al., 2019*). The purpose of R2VR is to implement and perform VR experiments, and record and analyse data for scientists while minimizing the need for different tools and expertise beyond the R language. We adopted a similar approach to that of *Loup et al. (2018)*, which allows non-VR developers to create VR experiences without the need for VR programming expertise. Their approach is based on the development of an open-access pipeline in which non-VR programmers can generate and use versatile VR scripts for their own purposes. The pipeline simplifies the development of VR environments by connecting game engines with VR assistance tools. Similarly, the R2VR package uses the WebXR to generate VR experiences for non-developers and to collect data from R. The technical challenges relate to (1) the ability for an R user to interact with a VR scene via WebSocket connections between R and a WebXR Device API (see R2VR description) and (2) the creation of a database to store and retrieve data from VR experiments, which, in the present case, is achieved via a Node API.

In this paper, we first describe the functions of R2VR to assist in the creation of VR experiments and its applications in environmental conservation research. We then present a comparative review of analogous studies from three different perspectives: the user, the developer and the quantitative ecologist. Following this, we present three case studies in which we have implemented the R2VR package. The paper concludes with a general discussion.

The R2VR package opens up many new directions of enquiry among quantitative ecologists and software developers. These include the elicitation of expert information, the analysis of elicited responses and the validation of these data. It is beyond the scope of this paper to discuss these issues, although we point the reader to (*Choy, O'Leary & Mengersen, 2009*; *Bednarz et al., 2016*; *Brown et al., 2016*; *Santos-Fernandez et al., 2020*; *Santos-Fernandez & Mengersen, 2020*) for further reading.

## R2VR DESCRIPTION

The R2VR package uses A-Frame (https://aframe.io/) and WebXR Device API platforms (https://www.w3.org/TR/webxr/) (Fig. 1) for building VR experiences. These are open-source and make the VR functionality accessible to people with basic knowledge in web programming (*Santos & Cardoso, 2019*). VR experiences are composed of assets (`a_asset`) that can be an image, texture or model; entities (`a_entity`) indicating the placeholder for an object; and scenes (`a_scene`) composed of all the created objects. The R2VR package uses the A-Frame architecture which allows VR scenes to be composed and served directly from

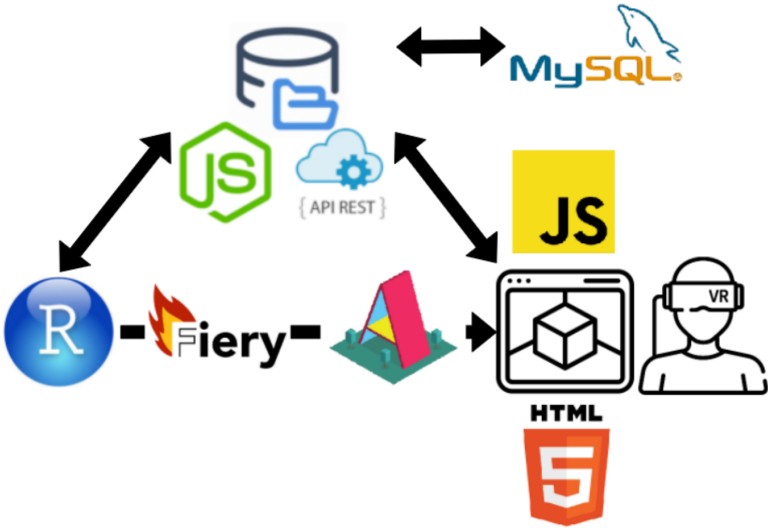

**Figure 1** **Workflow of the R2VR package.** A function package is used to start a Fiery server from the R console and render WebXR Device API scenes via harnessing Mozilla's A-Frame framework. This allows for the scene to be composed through the R interface and served into HTML and JavaScript which displays the VR scene in a WebXR environment (web browser and/or VR headset). There is a WebSocket connection between the Fiery server and the client which allows for R console commands to directly communicate with the user (e.g., display a question with the pop() function) in the VR environment. The recorded data is stored in an online MySQL database through a RESTful MVC NodeJS Application Programming Interface (APIRest). The Node API endpoints are made accessible for data fetching into R so all user responses can be analysed. There is an interoperable flow of data between R and VR through the implementation of the WebSocket and an API connections.

an R script. It then displays them in a web browser via a local IP (Internet Protocol). VR experiments are performed by typing functions directly into the R console. VR experiments can be performed from any device connected to the same network as R2VR, including smartphones, laptops and VR headsets (e.g., HTC Vive, Oculus Rift, Oculus Quest, Oculus Go, Google Daydream, Samsung GearVR and HTC Vive Focus). Once the VR scenes have been created data can be collected from users immersed in the scene, stored in an online database, and retrieved directly in R. The R2VR package does this via a RESTful Node.JS Application Programming Interface (APIRest, Fig. 1). Instructions on setting up the package and examples of VR scene creation using 360-degree images are given in the next section.

The R package is hosted by a Github repository: (https://github.com/ACEMS/r2vr) and can be installed using the command:
`devtools::install_github("ACEMS/r2vr")`

The package functionality is composed of five generic functions, which are shown in Table 1. The rendering of VR scenes in the web browser is started and stopped using the functions start() and stop(), respectively. The toggling of questions (on or off) is controlled by the pop() function and images are changed via the function go(). Answers are automatically saved within the online database hosted (https://www.db4free.net/). Data are retrieved using the function read(). From the users' point of view, there is no need

**Table 1 Description of the main functions included in the package.** See the help files for more details about the function arguments.

| Function | Description |
| --- | --- |
| start() | starts the VR server on the web browser |
| end() | kills the VR server |
| pop() | displays the question on the image |
| go() | jumps to another image |
| read() | retrieves the data from the database |

for external installation and manipulation since this is automatically done by the R2VR package. The data collected during an experiment are then curated and can be visualised in real time from the R console.

To ensure that the framework is fully operational, VR experiments were conducted using the R2VR package installed on local computers at Queensland University of Technology (QUT). Co-authors and several collaborators were immersed into three virtual ecosystems composed of four different 360-degree images and then asked to answer questions using a Samsung GearVR headset and Oculus Quest. The data collected during these experiments were used to create the visualisations in the R2VR package vignette. The "Interaction" vignette is included in the R2VR package and contains instructions on how to reproduce the case studies. It is also displayed in the Supplementary Material.

# COMPARATIVE STUDIES

In this section, we embed the R2VR package in the body of related literature. We first note the merit of VR compared with 2D technologies, then focus on a comparative review of other platforms. Noting the intended audience for R2VR, we address the latter from three perspectives: the user, the ecologist and the quantitative ecologist.

## VR versus 2D technologies

The benefits of using 3D technologies including VR and Augmented Reality (AR) compared to 2D environments have been widely demonstrated in the literature. *Akpan & Shanker (2019)* performed a meta-analysis to compare 3D and traditional 2D technologies and found enhanced performance and quality in 3D settings. VR technology has been found effective for training, gamification and collaboration; resulting in more suitable experiences and motivated participants (*Kavanagh et al., 2017*).

## R2VR versus other platforms

The purpose of R2VR is to provide a malleable tool that the scientific community can access to easily create their own VR environments that collect data. The tool is particularly targeted to the very large community of R users, noting that R is a very popular programming languages in the world with an open-access to many statistical packages for data analyses.

The package R2VR uses the A-Frame platform (https://aframe.io/) to create the VR environments and R to run the experiment and read the data. The A-Frame platform is a common choice for development of VR environments (*Dibbern et al., 2018*), although

**Table 2 Comparisons R2VR (A-Frame embedded R) and Unity.**

| | | R2VR | Unity |
|---|---|---|---|
| User | Pros | Accessibility, Run on the web, Compatible with most VR headsets | Mature, Ongoing development by large firm and massive community, Compatible with most VR headsets |
| | Cons | Not as mature or as commercially refined | App access, compatibility, and maintenance |
| Developer | Pros | Open-access sources, Relatively easy to implement, Accessible to the vast pool of web developers, Popular programming language | Flexible, Customizable, Extensive documentation and community, Easily integrated with other software, Mature tool support and high-quality, Integrated Developer Environment Tools, Asset Store resources are large and complete |
| | Cons | Background in web programming, Not as flexible | Very specific programming language(s), Complex environment, Need licence for research projects |
| Quantitative ecologist | Pros | Generic, Multipurpose, Use a unique programming language, Collect data in flexible format | Can produce refined user experiences for non-domain specialists |
| | Cons | Access to internet mandatory, Potential issues with free hosting provider | Specific purpose, Use of more than one platforms to perform experiments, Manipulate more than one programming language |

other 3D game engines such as Unity are popular for the integration of VR experiments within other research fields, including neuroscience (*Vasser et al., 2017*; *Jangraw et al., 2014*) and human behaviour (*Brookes et al., 2019*).

Here, we compare positive and negative characteristics of R2VR (A-Frame embedded in R) and Unity from the perspective of the user, the developer and the quantitative ecologist (Table 2). The user is defined as a participant in the VR experiments with the duty of answering interview questions. The developer is the programmer that focus on providing the immersive environments. We define the category quantitative ecologist as researchers that use VR experiments as a tool to answer research questions. They hold the responsibility of developing interview questions and immersive environments, performing experiments, and collecting data for analyses.

The comparison was based on two sources: first hand experience and published literature. The first hand experience was based on elicited information from users that participated in the ecological case studies (see below), developers in Web- and Unity- programming, and quantitative ecologists that conceptualized these studies. Information from the published literature was extracted using a systematic reading on which positive and negative characteristics of VR experiments in applied sciences are discussed by the authors. The published literature was based on key references cited in the following papers (*Dibbern et al., 2018*; *Nebeling & Speicher, 2018*; *Nguyen, Hite & Dang, 2018*; *Santos & Cardoso, 2019*).

R2VR provides the ability for researchers to customise the data retrieved from experiments which is made accessible into the statistical programming language, R. While this purpose may sounds similar in other studies, we did not find in the literature a tool that enables to create VR experiments, that is easy to use for non-programmers, that can generate data for use in data analyses, and is generic enough to be re-purposed. To our knowledge, existing VR packages for applied sciences require specialized hardware and substantial programming knowledge to customise the experiment beyond the case study for which the tool was developed. The tools that we found in the related literature

require downloading, setting up, and interfacing with Unity, an additional step that many ecologists wish to avoid. Some of these packages focus on customising 3d objects from models with pre-fabricated environments (*Vasser et al., 2017*; *Jangraw et al., 2014*; *Bexter & Kampa, 2020*), in comparison to R2VR which gives the freedom to change the environments, customise expected data responses, and interact with VR to change images or ask questions in real time. Some are customizable, but extensive knowledge in C# and Unity knowledge are needed to re-purpose them (*Brookes et al., 2019*).

## ECOLOGICAL CASE STUDIES

The R2VR package was used to replicate parts of VR experiments developed in previous studies. We used this approach to demonstrate stepwise how to build a VR environment directly from R. The content for reproducing these case studies is composed of twelve 360-degree images, R2VR functions and R scripts with interview questions. These resources are included with the R2VR package.

### Case study 1: koala

*Leigh et al. (2019)* developed a modelling framework for estimating the spatial distribution of koalas (*Phascolarctos cinereus*) in Southeast Queensland (SEQ), Australia. The model integrated koala sightings from volunteers, sightings estimated from thermal imagery collected using drones, and data from experts elicited using VR technology. Experts were immersed in 360-degree images of forested areas and asked about (1) the likelihood of koalas being present and (2) habitat suitability for koalas, with associated confidence in their estimates. Answers were manually recorded and transferred to CSV files. Probabilities obtained from elicitation were modeled using a beta regression and subsequently integrated with presence-absence data obtained from volunteers and thermal images within logistic regression models. The results demonstrated that incorporating data elicited using VR into the statistical models produced better predictions of koala distribution and better characterisation of their habitats.

For the purpose of the present study, co-authors and collaborators were immersed in a sample of these 360-degree images of forested areas and were asked: "Do you see a koala?". They responded by selecting "yes" or "no" within the VR scenes (Fig. 2A). The associated data table *koala* is composed of five variables and populated with the user's responses (Table 3). User names and answers were retrieved by using the starting and ending times of the elicitation (*recordedOn* in Table 3) in the R script. VR scenes were restarted for each participant between case studies.

### Case study 2: jaguar

A team of QUT researchers and collaborators conducted a study in Peru to help the development of a jaguar conservation corridor across South America (*Zeller et al., 2013*). Part of the study involved modelling the distribution of jaguars (*Panthera onca*) using jaguar sightings from the Shipibo and Conibo indigenous communities. *Mengersen et al. (2017)* considered several occupancy and abundance models, which included environmental covariates to account for uncertainty associated with different types of jaguar sightings (e.g.,

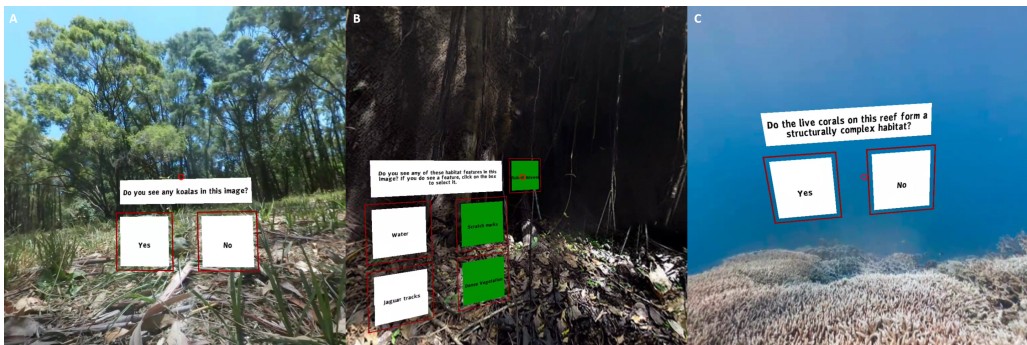

**Figure 2** **Case studies developed using the R2VR package with (A) Koala, (B) Jaguar and (C) Coral reef studies.** The screenshots show the questions that were asked as part of the framework testing. Coral reef images were provided by Underwater Earth / XL Catlin Seaview Survey / Christophe Bailhache. Short videos of the virtual reality scenes can be seen at: https://youtu.be/el08HKysZX8.

**Table 3** **Description of the data obtained from the elicitation stored online in table *koala*.**

| Variable | Description |
|---|---|
| *id* | classification id |
| *image_id* | unique identifier of the image |
| *image_file* | image's file name |
| *binary_response* | 0 (absence) or 1 (presence) |
| *recordedOn* | date-time of the classification event |

visual sighting, vocalisation, tracks, etc.) elicited from community members. The elicited information was then combined with visual and sound recordings to create immersive environments of the Peruvian jungle (http://vis.stats.technology/) and interview jaguar experts on several aspects of jaguar habitat (*Bednarz et al., 2016*).

Here, we used the R2VR package to show four 360-degrees images of the Peruvian jungle. Co-authors and collaborators were asked to consider characteristics known to affect jaguar habitat suitability, such as the presence/absence of water, jaguar tracks, jaguar scratches on trees, and dense vegetation. They could select more than one indicator by clicking on the associated boxes within the VR scenes (Fig. 2B). The *jaguar* data table (Fig. S1) is composed of eight variables (four relating to presence/absence of indicators, four for metadata, *id*, *image_id*, *image_file* and, *recordedOn* - shown in Table 3).

## Case study 3: coral reef

The reef VR experiments was originally developed to estimate indicators of coral reef aesthetics based on people's perception of reef beauty (*Vercelloni et al., 2018*). Three groups of people (marine scientists, experienced divers and the general public) were interviewed while immersed in 360-degree images of the Great Barrier Reef, Australia. The presence/absence of variables that represent different ecological characteristics of coral reefs and their opinions about reef beauty were used to parameterize a logistic regression model in order to gain knowledge about what makes a coral reef beautiful. The model results

suggested that a structurally complex reef with diverse colours had a positive influence on reef aesthetics.

We asked co-authors and collaborators to look at four virtual reefs and answer the question "Do the live corals on the reef form structurally complex habitats?" (Fig. 2C). After the classification of the images, we obtained a data table called *coral reef* (Fig. S1) composed of five variables with binary responses (0 for non-complex reef and 1 for complex reef) to the question.

## DISCUSSION

The package R2VR offers open access to VR experiments for the scientific community. It supports the development of new research tools by creating a more complex environment for participants and lowers the technical barriers for ecologists by easing the learning curve for VR programming and uptake VR technologies.

There are several advantages to embedding VR experiments in R. Connecting VR experimental results directly to R allows researchers to process, manipulate and visualize data, and access to the latest statistical methods in ecology. The generic, multipurpose and unique programming language of R2VR is key to increasing the uptake of VR as an accessible research tool for ecologists and other applied scientists. Previous efforts to simplify VR development for non-programmers allowed the integration of VR in different research fields (*Jangraw et al., 2014*; *Vasser et al., 2017*; *Brookes et al., 2019*) but those remained focused on specific purposes and are not easily adaptable to other research questions. Ecologists have different needs, including the flexibility to modify VR scenes and collect data for analyses in a language that they can understand. These needs are different from users' and developers' perspectives.

The current implementation of R2VR uses A-Frame to create VR environments written in the R language. Another R package "shinyframe" uses this combination to visualise 3D plots (https://cran.r-project.org/web/packages/shinyaframe/shinyaframe.pdf). Further developments could include using Unity instead of A-Frame within the R2VR package. Similarly, VR environments could be coded in a different language other than R but still familiar to the ecologists. For example, R2VR could be rewritten for Python (https://www.python.org/) which is an even more popular language among applied scientists (https://www.tiobe.com/tiobe-index/), but not as developed in terms of statistical capabilities. The availability of such tools will greatly help to fulfill the need of ecologists and ultimately increase the adoption of immersive experiments in applied science.

It is increasingly being shown that VR technology facilitates the public involvement in environmental conservation by creating engaging learning environments (*Queiroz et al., 2019*; *Fauville, Queiroz & Bailenson, 2020*). R2VR provides a fundamental framework for citizen science projects that could use VR to perform different activities including online data collection, data extraction from existing records, and knowledge sharing (*Mazumdar et al., 2018*). It can also facilitate a natural link between VR experiments and citizen science projects by offering an open-access tool for research scientists to build their own VR environments. In this way, members of the public can perform tasks in the same perceptual

environment as might an ecologist and collect useful data. The web framework that is associated with the R2VR package means that online citizen science projects could be developed at low cost. However, modifications of the R2VR server (from local to web server) and the automation of package functions are required to support this goal.

To date, R2VR has only been used to elicit information from static 360-degree images without audio. However, the A-Frame software offers additional VR experiences, such as the inclusion of soundscapes (https://www.8thwall.com/playground/aframe-audio-and-media-recorder) and 3D mesh from Geographical Information System layers (https://milesmcbain.xyz/posts/r2vr3-shading-meshes-in-webvr/) that could be easily integrated into the R2VR workflow. While this development will offer a greater level of virtual immersion, further research and development is required to understand how to increase the knowledge gained from VR experiments (*Fauville, Queiroz & Bailenson, 2020*). By having the capability to design their own experiments or being interviewed, experts in ecology may enhance the potential of VR to support new scientific discoveries due to the priming of visual memories from known environments and ecological knowledge (*Brown et al., 2016*; *Vercelloni et al., 2018*).

Further package developments will also enhance the access and security of the database. The current db4free database connected to R2VR is a free hosting provider. In this implementation we used db4free to avoid the payments associated with data hosting provider and the burden of installing a local database, but we acknowledge that this choice may cause other issues. We suggest that users check the db4free website to ensure that it is a suitable host provider for their experiments, locate the codes within the package that connect to the online database and modify them if necessary. We also recommend that they regularly save their data on their own machine using the `read()` function from the R2VR package and `write.csv()` or `save()` functions from R. Additional security improvements would include adding authentication/authorization to secure API endpoints. Whilst the Node server is using HTTPS, the R server is currently using the HTTP protocol. The current implementation contains anonymous and non-sensitive data. However, improvements to the Fiery server's security may be beneficial for use cases with non-anonymous sensitive data. Another development goal is to support the creation of more flexible data tables.

## CONCLUSIONS

In conclusion, we have demonstrated that it is now possible to create VR experiments in a seamless statistical programming environment that is highly popular and hence accessible among ecologists. This tool offers new horizons for ecological research as data generated from VR experiments can be used by researchers themselves, but might also be integrated with information collected by other technologies. This provides a new tool for filling in data gaps in ecosystems with poor data density or coverage, and allowing for a better understanding of ecological systems. R2VR is also applicable to other ecosystems as well as directly generalisable to non-ecological VR experiments.

## ACKNOWLEDGEMENTS

We thank our collaborators on the VR projects: Ross Brown and Allan James for their insights about the comparison Unity engines and A-Frame and, Ella Wilson, Taylor Gregory, Samuel Clifford, Catherine Leigh, Bryce Christensen, Tomasz Bednarz, June Kim, Alan Pearse, Gavin Winter, Jacinta Holloway, Jacqueline Davis, Vanessa Hunter and Kevin Burrage for their contribution to the original studies.

### Funding

Kerrie Mengersen received support from her Australian Research Council Laureate Fellowship (ID: FL150100150). The funders had no role in study design, data collection and analysis, decision to publish, or preparation of the manuscript.

### Grant Disclosures

The following grant information was disclosed by the authors:
Australian Research Council Laureate Fellowship: FL150100150.

### Competing Interests

The authors declare there are no competing interests.

### Author Contributions

- Julie Vercelloni and Edgar Santos-Fernandez conceived and designed the experiments, performed the experiments, analyzed the data, prepared figures and/or tables, authored or reviewed drafts of the paper, and approved the final draft.
- Jon Peppinck conceived and designed the experiments, performed the experiments, performed the computation work, prepared figures and/or tables, authored or reviewed drafts of the paper, and approved the final draft.
- Miles McBain conceived and designed the experiments, performed the computation work, authored or reviewed drafts of the paper, and approved the final draft.
- Grace Heron performed the experiments, analyzed the data, authored or reviewed drafts of the paper, and approved the final draft.
- Tanya Dodgen performed the experiments, authored or reviewed drafts of the paper, and approved the final draft.
- Erin E. Peterson and Kerrie Mengersen conceived and designed the experiments, authored or reviewed drafts of the paper, and approved the final draft.

### Data Availability

   Raw data and code are available at GitHub: https://github.com/ACEMS/r2vr.

### Supplemental Information

Supplemental information for this article can be found online at http://dx.doi.org/10.7717/peerj-cs.544#supplemental-information.

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
