# Peer review of "Connecting virtual reality and ecology: a new tool to run seamless immersive experiments in R"

_PeerJ Computer Science, doi:10.7717/peerj-cs.544_

## Round 0.1 · original submission · Major Revisions

Please pay attention to all reviewers' comments but particularly the comments from reviewer 2, who has raised some serious concerns across the article sections.

Also, in your response letter, please make sure to highlight how this paper falls into the aims and scope of PeerJ.

PeerJ website (https://peerj.com/about/aims-and-scope/cs) states that "PeerJ Computer Science only considers Research Articles and Literature Review Articles. It does not accept Hypothesis Papers, Commentaries, Opinion Pieces, Case Studies, Case Reports etc". Please also state in which category your paper falls.

·

Basic reporting

This article is very clear thanks to its good structure and the right expressions used. Literature references used have a quantity adapted to this paper. The majority of the articles cited are very recent and from reputable publishers.
PeerJ recommends a title between 20 and 60 words to be more concise. In your case, it might be interesting to add some details. While reading the current title, the reader might think of a solution to see the results of R in a virtual reality headset.

Experimental design

The research question is well defined in the introduction section: Make the design of virtual reality environments more accessible to R users without virtual reality skills. In addition, the choices of each r2vr module have been justified to ensure that this solution does not add any extra budget to the targeted users. The major advantage of this approach is the availability of the source code and the various resources that allow this package to be reused in another context.
A-Frame has many differences with other solutions such as Unity3d and Unreal. Both development environments also try to be accessible to as many designers as possible. Unity is the leader in virtual reality applications and Unreal Engine offers a Blueprint interface to be more accessible to novices (Dickson, 2017). It would therefore be preferable to add some justification for your choice of A-Frame for ecologists.

Validity of the findings

The section dedicated to the results gives us a better understanding of the case studies mentioned. This essentially describes the final product and its context.
Ideally, we would have liked to know the significant differences for designers or end-users with another solution. If no measurement was possible, then it would be preferable to detail your observations on r2vr's designers in these 3 cases.
At the end of the discussion, advice on how to use it is given. Improvements and problems concerning the security and confidentiality of data storage could be discussed.

Additional comments

This well-written article proposes an innovative and accessible solution for a whole community. Three applications described have been created using the r2vr package. However, it would be interesting to define the criteria on which the situation has improved in order to facilitate experimentation by ecologists.

·

Basic reporting

The manuscript is well written and professional, and I enjoyed reading it.
However, there are some key issues with it in it's current form.
For example, the background section begins with a very narrow interpretaion of virtual reality, namely that of an audiovisual system substituting for real world sensory input.
This description is not adhered to throughout the manuscript: audio is never mentioned again.
I checked the supplementary material and the videos were silent there too.
A further problem with this interpretation is it fails to account for dynamic interaction in VR, in particular where users may move and interact with the environment, with such actions resulting in an enhanced sense of presence [1] and a more immersive, 'vivid' experience.
The authors should be more critical of their interpretation and aware of the limitations this imposes on how others may find their system useful.

The second issue I have is the lack of a clear contribution behind this work.
Many prior work has tackled the issue of simplifying work flows for non programmers/experts so they may engage with VR technology (for example, see [2], [3], and [4]).
Its unclear how r2vr differs from these other attempts both in its implementation and its goal: there is overlap at a high level between psychologists, social sciences, and applied sciences (as in this work) to elicit responses from participants while immersed in a virtual environment.
Therefore I question the utility of r2vr over past attempts/existing solutions.
Furthermore, no comparison is made between r2vr and other systems: I recommend the authors perform a feature comparison to help convey *why* their system is both needed and *how* it compares to, or better yet improves upon, prior art/existing solutions.
This being said I very much welcome the open source nature of r2vr!
It is also worth considering the premise behind elicitation within versus outside of virtual environments.
This is not a trivial matter: please see [5] for discussion on administering questionnaires inside virtual environments.

My final issue revolves around the utility of the case studies.
Insufficient detail is provided, making it difficult to understand the point of their inclusion.
There is no discussion regarding how r2vr was perceived as a benefit (again, no comparison to other workflows/tools are provided), and also all 3 case studies were performed by the same research group involved in developing r2vr.
It is best to trial the system with other researchers not involved in the development/production of r2vr, and perform some qualitative analysis, perhaps involving lengthy debrief interviews with other researchers to get their perspsective and opinion on r2vr as a useful tool to add to their repertoire.
I would like further discussion in the paper around the problems of accessibility and non-power user (programmer) use, including how the authors intend to tackle this problem.

[1]\: Slater, M. (2009). Place illusion and plausibility can lead to realistic behaviour in immersive virtual environments. Philosophical Transactions of the Royal Society B: Biological Sciences, 364(1535), 3549–3557. https://doi.org/10.1098/rstb.2009.0138

[2]\: Vasser, M., Kängsepp, M., Magomedkerimov, M., Kilvits, K., Stafinjak, V., Kivisik, T., Vicente, R., & Aru, J. (2017). VREX: an open-source toolbox for creating 3D virtual reality experiments. BMC Psychology, 5(1), 4. https://doi.org/10.1186/s40359-017-0173-4

[3]\: Jangraw, D. C., Johri, A., Gribetz, M., & Sajda, P. (2014). NEDE: An open-source scripting suite for developing experiments in 3D virtual environments. Journal of Neuroscience Methods, 235, 245–251. https://doi.org/10.1016/j.jneumeth.2014.06.033

[4]\: Brookes, J., Warburton, M., Alghadier, M., Mon-Williams, M., & Mushtaq, F. (2020). Studying human behavior with virtual reality: The Unity Experiment Framework. Behavior Research Methods, 52(2), 455–463. https://doi.org/10.3758/s13428-019-01242-0

[5]\: Regal, G., Voigt-Antons, J.-N., Schmidt, S., Schrammel, J., Kojić, T., Tscheligi, M., & Möller, S. (2019). Questionnaires embedded in virtual environments: Reliability and positioning of rating scales in virtual environments. Quality and User Experience, 4(1), 5. https://doi.org/10.1007/s41233-019-0029-1

Experimental design

No experiment is provided: instead, the authors have provided 3 case studies all of which involved the use of r2vr.
Please see the previous section 'Basic Reporting'.

Validity of the findings

It is unclear what the 'findings' from this work are.
The material is quite light, giving a brief business case for r2vr, offering a high level overview of the system design, and then finally 3 case studies involving its use.
The authors conclude that they 'have demonstrated that it is now possible to create VR experiments via a unique software', but as I have pointed out in the 'Basic Reporting' section, the scientific community already knew this.

Additional comments

There is a typo on line 45, and line 198.
Please consider using footnotes for URLs.
The link provided in line 197 returns a 404.

---

## Round 0.2 · Minor Revisions

Please pay attention to the comments of reviewer 2. Also, highlight your response to my previous comments more explicitly as you did for reviewers 1 & 2.

·

Basic reporting

The authors have taken into account my suggestions.

Experimental design

The authors have taken into account my suggestions.

Validity of the findings

The authors have taken into account my suggestions.

Additional comments

This well-written article proposes an innovative solution that is accessible to a whole community. The three applications described were created using the r2vr solution. These case studies, together with the source code made available, demonstrate the interest that should be shown in this new method.

·

Basic reporting

L45 to 52: In my review I criticized the narrow interpretation of VR. To say many current VR research applications use only a subset of features, meaning multisensory stimuli, is an unfounded statement. For example, a recent survey paper concluded 84% of application use at least one additional sensory modality (haptics) to vision [1]. Please remove this statement or establish a basis for it.

L144 to 171, 225 to 240, and Table 2: These additions are very welcome. However, they require further refinement. For example, the authors make the case of the existing tools requiring knowledge of C# and Unity. This would be a valid stance to take if it weren't for the fact that the authors then discuss the use of technologies and languages beyond base R. Based on my understanding of the r2vr framework--so please correct me if I am wrong--for end users to do anything more than pop-up existing assets in a VR environment, they would need to familiarize themselves with other technologies and languages e.g., python, HTML. In my original review, my point was for the authors to state clearly "convey *why* their system is both needed and *how* it compares to, or better yet improves upon, prior art/existing solutions." Based on my reading of this rebuttal, the business case behind r2vr is to encourage uptake of VR technologies by ecologists through the stratgey of easing the learning curve e.g., non-power users I referred to in my original review. If this is the case, then the authors should state this clearly in the manuscript. With respect to Table 2, there is a lot of information justifying its inclusion in the rebuttal that is missing from the proposed changes to the manuscript e.g., the first paragraph in gray font on page 6 of the rebuttal. Can the authors please include this detail in the manuscript along with detail regarding how pros and cons were elicited from the literature?

[1]: Melo, M., Gonçalves, G., Monteiro, P., Coelho, H., Vasconcelos-Raposo, J., & Bessa, M. (2020). Do Multisensory stimuli benefit the virtual reality experience? A systematic review. IEEE Transactions on Visualization and Computer Graphics, 1–1. https://doi.org/10.1109/TVCG.2020.3010088

Experimental design

No comments other than what is above

Validity of the findings

No comments other than what is above

---

## Round 0.3 · accepted · Accept

Thank you for revising the article considering reviewers' comments.

·

Basic reporting

The authors have addressed my previous comments. I have no more.

Experimental design

The authors have addressed my previous comments. I have no more.

Validity of the findings

The authors have addressed my previous comments. I have no more.